# Effect of Thoracic Gas Volume Changes on Body Composition Assessed by Air Displacement Plethysmography after Rapid Weight Loss and Regain in Elite Collegiate Wrestlers

**DOI:** 10.3390/sports7020048

**Published:** 2019-02-19

**Authors:** Emi Kondo, Keisuke Shiose, Yosuke Yamada, Takuya Osawa, Hiroyuki Sagayama, Keiko Motonaga, Shiori Ouchi, Akiko Kamei, Kohei Nakajima, Hideyuki Takahashi, Koji Okamura

**Affiliations:** 1Japan Institute of Sports Sciences, 3-15-1, Nishigaoka, Kita-ku, Tokyo 115-0056, Japan; hiroyuki.sagayama@jpnsport.go.jp (H.S.); keiko.motonaga@jpnsport.go.jp (K.M.); ouchi.shiori@gmail.com (S.O.); akiko.kamei@jpnsport.go.jp (A.K.); kohei.nakajima@jpnsport.go.jp (K.N.); hideyuki.takahashi@jpnsport.go.jp (H.T.); 2Department Faculty of Sports and Health Sciences, Fukuoka University, 8-19-1 Nanakuma, Jonan-ku, Fukuoka 814-0180, Japan; shiose@fukuoka-u.ac.jp; 3Section of Healthy Longevity Research, National Institute of Health and Nutrition, National Institutes of Biomedical Innovation, Health and Nutrition, 1-23-1 Toyama, Shinjuku-ku, Tokyo 162-8636, Japan; yamaday@nibiohn.go.jp; 4Department of Sports Wellness Sciences, Japan Women’s College of Physical Education, Tokyo 157-0061, Japan; tack_415@yahoo.co.jp; 5Japan Society for the Promotion of Science, Kojimahi Business Center Building, 5-3-1, Kojimachi, Chiyoda-ku, Tokyo 102-0083, Japan; 6Graduate School of Sport Sciences, Osaka University of Health and Sport Sciences, Osaka 590-0496, Japan; okamura@ouhs.ac.jp

**Keywords:** body composition, thoracic gas volume, rapid weight loss, weight regain, weight-categorized athletes

## Abstract

We investigated the effect of rapid weight loss (RWL) and weight regain (WR) on thoracic gas volume (V_TG_) and body composition assessment using air displacement plethysmography (ADP) in male wrestlers. Eight male elite collegiate wrestlers completed a RWL regimen (6% of body mass) over a 53-h period, which was followed by a 13-h WR period. ADP was used at three time points (baseline (T1), post-RWL (T2) and post-WR (T3)) according to the manufacturer’s testing recommendations. The total body water and bone mineral content were estimated using the stable isotope dilution method and dual energy X-ray absorptiometry, respectively, at the same time points. Body composition was assessed with two-component (2C) or four-component (4C) models using either the measured V_TG_ (mV_TG_) or predicted V_TG_ (pV_TG_). Measured V_TG_ increased from T1 to T2 (0.36 ± 0.31 L, *p* < 0.05) and decreased from T2 to T3 (−0.29 ± 0.15 L, *p* < 0.01). However, the changes in fat mass and fat free mass, which were calculated by both 2C and 4C models, were not significantly different when compared between calculations using mV_TG_ and those using pV_TG_. Our results indicate that V_TG_ significantly changes during RWL and WR, but both measured and predicted V_TG_ can be used to assess changes in body composition during RWL and WR.

## 1. Introduction

Weight-categorized sports athletes, such as those participating in wrestling, judo and boxing, often reduce their weight via food/fluid restriction, saunas and training in rubber/plastic suits. After being weighed, they regain their weight rapidly by consuming food/fluids in order to obtain a physical or mental advantage over their opponent(s) before competitions [1,2,3]. However, severe rapid weight loss (RWL; >5% of body mass (BM)) over the course of just one week induces dehydration [4,5], heatstroke [6] and in the most severe cases, even death [7]. Therefore, it is important to accurately identify specific changes in their body composition associated with RWL and weight regain (WR) in order to form a strategy for weight management.

Body density (Db) is one of the most important variables used to estimate body composition in either two-component (2C) or multi-component models (e.g., three- or four-component (4C) models). Currently, Db is often estimated by air displacement plethysmography (ADP), which estimates body volume (BV) and thoracic gas volume (V_TG_) before computing Db using body volume and BM. The volume of the ADP chamber is estimated using Boyle’s law. However, it is important to note that the air in the lungs and the respiratory tract is 40% more compressible than the adiabatic air [8]. Thus, to measure BV with adequate accuracy, it is necessary to either eliminate or account for the effects of lung and respiratory tract gas volume [9]. The V_TG_ is measured using a breathing circuit internal to the system [10]. However, it is occasionally not possible to measure this in a clinical setting. V_TG_ is usually regarded as being unchanged, even if BM changes. Therefore, a fixed V_TG_ (e.g., predicted or previously measured V_TG_) is usually used in sports science [11].

In contrast, Minderico et al. reported that V_TG_ increased significantly after a 16-month weight loss program in overweight or obese women [12]. This increased V_TG_ may have been due to the diaphragm being able to expand more into the abdominal cavity due to the reduction in abdominal fat. In their study, the fat mass (FM) loss using the predicted V_TG_ was greater (by 400 g) than that using the measured V_TG_. It is important to note that if V_TG_ changes in the athletes during weight loss and regain, the associated change in V_TG_ would not be trivial.

With these facts in mind, the purpose of this study was to investigate the effect of RWL and WR on V_TG_, which can affect the accuracy of body composition assessment in male wrestlers via ADP. To confirm the accuracy of ADP, we compared the 2C model with the 4C model, which is regarded as the gold standard for body composition assessment. A previous study [13] reported that the cross-sectional visceral area decreased by ~25.8% when wrestlers lost ~7.3% of their BM. In this situation, V_TG_ may increase due to the diaphragm being able to expand more into the abdominal cavity. The hypothesis of the present study was that V_TG_ would increase during RWL and decrease during WR, which could lead to a difference in the estimation of body composition changes between values that were calculated with the measured and predicted V_TG_.

## 2. Materials and Methods

### 2.1. Participants, Ethical Approval and Consent to Participate

The study was conducted from January to March (winter season), which was the offseason of wresting. Subjects from two Kanto area teams in Japan were informed about this study by their coaches and the players who volunteered to cooperate and met the study criteria were recruited. This study involved ten male collegiate wrestlers (mean age: 20.4 ± 0.5 years; mean height: 168.4 ± 4.2 cm; mean BM: 73.0 ± 7.9 kg) who competed in international or national tournaments. All participants were over 18 years of age, had experienced losing > 6% of their BM before a major competition and did not have any metabolic, thyroid or heart diseases. Two subjects were excluded because they did not fulfill the criteria required for V_TG_ measurement during ADP measurement (merit value ≤ 0.50 Hz and airway pressure < 35.0 cm H_2_O) at least once during three (baseline, T1; post-RWL, T2; and post-WR, T3) measurements. All subjects signed appropriate informed consent forms before being evaluated. This study was conducted in accordance with the declaration of Helsinki and all procedures were approved by the institutional review board at the Japan Institute of Sports Sciences (approval number 036 in 2014).

### 2.2. Procedure

The protocol was developed based on previous studies [14,15]. All participants refrained from taking alcohol or stimulant beverages and were given instructions to consume meals, snacks and beverages as usual for at least 24 h prior to T1 measurement. They were instructed not to exercise for at least 12 h prior to T1 measurement and to abstain from consuming food/fluids, except for water, after 23:00 on the night before the measurement.

We obtained anthropometric measurements (described below) and determined the body composition of all participants at 06:30 (T1). After the measurements were obtained, they were asked to lose 6% of their BM using their own methods, such as reducing food/fluid intake or sweating using saunas/training with plastic/rubberized suits under the supervision of their coaches at home or their training facility. They were instructed to come to the laboratory for T2 measurements 53 h after the T1 measurement. After the T2 measurements were taken, the participants were provided with a prescribed diet between 17:30 and 23:00 (WR period). The prescribed diet was designed based on a pilot study conducted in our laboratory that determined ad libitum consumption for 5 h after RWL. On the next morning (06:30), T3 measurement was performed.

### 2.3. Anthropometric Data and Air Displacement Plethysmography

Height was measured barefoot to the nearest 0.1 cm with a stadiometer (A&D Co. Ltd, Tokyo, Japan). The participants were weighed to the nearest 0.01 kg after voiding of their bladders when they wore a bathing suit and swim cap and stood barefoot on an electronic scale connected to the plethysmograph computer (BOD POD, COSMED Inc., Rome, Italy). The participant’s body volume (BVraw) and V_TG_ were assessed by ADP using BOD POD software (version 4.24, COSMED Inc., Rome, Italy), according to the manufacturer’s testing recommendations and guidelines [8], in a room with a controlled temperature of 23 °C. All participants practiced the steps needed fir V_TG_ measurement prior to the baseline measurement by using the equipment in practice mode until they passed the predetermined criteria (merit value ≤ 0.50 Hz and airways < 35.0 cm H_2_O), which were tougher than previous standards [16] in order to obtain more accurate measurements. The merit of the V_TG_ measurements, which indicates how two curves (airway pressure and chamber pressure) fit into their sum of the square, was demonstrated by the device and did not differ across the three measurements (baseline: 0.04 ± 0.06 Hz; post-RWL: 0.03 ± 0.05 Hz; post-WR 0.07 ± 0.14 Hz) (a value equivalent to zero indicates perfect agreement) [8]. The airway pressure, as shown by the device, also did not differ across the three measurements (baseline: 17.6 ± 6.9 cm H_2_O; post-RWL: 19.2 ± 7.2 cm H_2_O; post-WR: 17.5 ± 8.5 cm H_2_O). The interindividual and intraindividual standard deviations (SD) of V_TG_ based on five measurements obtained from eight participants at T1 were 0.71 L and 0.16 L, respectively.

The participants’ Db, BV and V_TG_ values were calculated as previously reported [8] using Equations (1)–(3).
Db (kg/L) = BM (kg)/BV (L)(1)
BV (L) = BVraw + 0.40 × V_TG_ − Surface area artefact (L)(2)
Surface area artefact (L) = k × Body surface area (cm^2^)(3)

The surface area artefact consisted of the body surface area [17] and the constant k (−4.671 × 10^−5^).
Body surface area (cm^2^) = 71.84 × BM (kg)^0.425^ × Height (cm)^0.725^(4)
Measured V_TG_ (L) = (m/1.4) − dead space of the breathing apparatus(5)
where m was derived from the least squares solution of (Yi − (mXi + b)), with Yi equal to the product of the chamber pressure and chamber volume and Xi equal to the corresponding airway pressure during the 3-s occlusion.

The predicted V_TG_ was estimated from the functional residual capacity (FRC) as shown in Equation (6).
Predicted V_TG_ (L) = FRC (L) + 0.5 × VT(6)
where VT was the tidal volume estimated during computer-guided breathing. The predicted FRC was estimated using Equation (7) [18]:FRCpred (L) = 0.0472 × Height + 0.0090 × Age − 5.290(7)

The proportion (%) of fat mass (%FM) in each participant was calculated using Equation (8) [19]:Percentage of FM = (4.95/Db − 4.50) × 100(8)

After this, each participant’s fat mass (FM) and fat free mass (FFM) values were calculated using %FM and BM, as shown in Equations (9) and (10):
FM (kg) = BM (kg) × %FM/100(9)
FFM (kg) = BM (kg) − FM (kg)(10)

Based on the data obtained in a previous study from twenty-two subjects, the coefficient of variation (CV) in our laboratory for BV and %fat calculation was 0.1% and 4%, respectively [20]. The intraindividual SD in a subgroup of athletes (n = 13; age, 21 ± 3 years; height, 170.0 ± 5.7 cm) was 0.16 L for mV_TG_ and 0.43 kg for FM for 2 consecutive days.

### 2.4. Total Body Water

Each participant drank ~0.12 g of ^2^H_2_O (^2^H_2_O 99.9 atom%; Taiyo Nippon Sanso, Tokyo, Japan) and ~1.5 g of H_2_^18^O (H_2_^18^O 20.0 atom%; Taiyo Nippon Sanso, Tokyo, Japan) per kg of their predicted total body water (TBW) one week before T1. After this, they drank ~0.06 g of ^2^H_2_O and ~0.75 g of H_2_^18^O per kilogram of their predicted TBW at T1, T2 and T3. The TBW was predicted to be 60% of initial BM. The bottle was rinsed twice with 30 mL of water. Each participant drank 220 mL of bottled water within 3 h of dosing and emptied their bladders 2 h after dosing. Blood samples were collected and aliquoted into Na_2_EDTA-containing tubes before dosing, which was repeated at 3 and 4 h after dosing. Blood samples were centrifuged at 4 °C for 10 min at 3000 rpm before being stored at −30 °C for subsequent isotope ratio mass spectrometry (IRMS) analysis (Hydra 20-20 Stable Isotope Mass Spectrometers; Sercon Ltd., Crewe, UK). TBW volume was calculated from the plasma concentration of ^2^H and ^18^O according to the plateau method [21]. We adjusted for the coefficient of ^2^H (1.041) and ^18^O (1.007) space, because these isotopes enter other pools within the body and exchange with non-aqueous components [22,23].

### 2.5. Dual-Energy X-ray Absorptiometry

Bone mineral content (Mo) and body composition were determined with whole-body scans by Dual energy X-ray absorptiometry (DXA; QDR 4500, Discovery A (S/N 84498) using a fan-beam scanner and software version 12.7.3.2 (Hologic, Waltham, MA, USA)). Wearing a swimsuit free of metal and/or plastic, each subject kept their body in the supine position for 5–10 min, according to the DXA protocol described by the manufacturer. The same technician positioned the participants, performed the scans and analyzed body composition, according to the manufacturer’s guidelines.

### 2.6. Four-Component Model of Body Composition

The 4C model used BM, BV [24] and TBW to calculate FM using Heymsfield’s equation [25], as described below:FM = (2.513 × BV) − (0.739 × TBW) + (0.947 × Mo) − (1.79 × BM)(11)

### 2.7. Energy and Macronutrient Intake

A survey of all food and fluid intake was conducted for 3 days, including two training days and one day off before T1 measurement (baseline), and during the 53-h RWL period. The participants were given scales and were instructed to weigh all consumed food, supplements and beverages in addition to taking a photograph with a ruler using a provided digital camera. A well-trained registered dietician calculated the nutrient intake from the diet records and photographs [26]. All diet records were analyzed using a computerized nutrient analysis program (Excel Eiyou-kun Version 6.0, Japan Food Composition Table Version 5, Kenpakusha, Tokyo, Japan). At baseline, it was calculated based on 3-day food records adjusted by an intake per day for 6 training days and 1 training-off day. The total intake in RWL period and WR period was over a period of 53 h and 5 h, respectively (Table 1).

### 2.8. Statistical Analysis

All data are expressed as means ± standard deviation (SD) along with 95% confidence intervals (CI). Data analysis involving BM was conducted using one-way repeated-measures analysis of variance (ANOVA). When a significant difference was detected, a multiple comparison test was performed using post hoc Bonferroni correction. The changes in V_TG_, BV, Db, FM and FFM over time and the differences associated with the mode of V_TG_ assessment (measured or predicted), were assessed using factorial repeated measures ANOVA with “time” and “methods” as the independent variables. The assumption of sphericity was tested using Mauchly’s test and if violated, the degrees of freedom were corrected using Greenhouse−Geisser estimates of sphericity. Post hoc analysis was conducted to identify differences between the means using a Bonferroni correction. To compare group means relating to the amount of change in V_TG_, BV and Db, we used factorial repeated measures ANOVA with “time” and “mode” as the independent variables. The means of the change in FM and FFM were compared by repeated one-way ANOVA and Bonferroni post hoc test. Bland−Altman plots were also constructed to display the errors and the bias between individual participants due to the difference in V_TG_ mode during V_TG_, FM and FFM calculation. All analyses were performed using SPSS version 24.0 (IBM Japan, Tokyo, Japan) and statistical significance was set at α < 0.05.

## 3. Results

The characteristics of eight participants at T1, T2 and T3 are presented in Table 2. The participants reduced their BM by 6.4 ± 0.5% from T1 to T2 (*p* < 0.001) and regained some of it from T2 to T3 (*p* < 0.001), although the average BM remained lower than that at T1 by 2.5 ± 0.7% (*p* < 0.001). TBW was significantly decreased between T1 to T2 (*p* < 0.001; 95% CI = −3.981 to −2.719) and increased significantly between T2 to T3 (*p* < 0.001; 95% CI = 2.484–3.816). Food weight/energy and macronutrient intake during the RWL period are described in Table 1. A Bland−Altman plot indicated that the difference in V_TG_ between the methodologies significantly increased in a linear manner with increasing V_TG_ during each measurement (T1, R^2^, 0.83, *p* < 0.01; T2, R^2^, 0.81, *p* < 0.01; T3, R^2^, 0.80, *p* < 0.01; Figure 1).

There was a significant interaction between time and mode in V_TG_ (*p* = 0.004), BV (*p* = 0.004) and Db (*p* < 0.001) (Table 2). V_TG_, BV and Db were not significantly different when compared between calculations using mV_TG_ and pV_TG_ at all time points. mV_TG_ increased significantly from T1 to T2 (*p* = 0.018; 95% CI = −0.722 to −0.078) and decreased from T2 to T3 (*p* = 0.003; 95% CI = 0.127–0.465), whereas pV_TG_ did not change over time. Therefore, the change in mV_TG_ was greater than that in pV_TG_ from T1 to T2 (*p* = 0.006; 95% CI = 0.158–0.642) and from T2 to T3 (*p* = 0.001; 95% CI = −0.426 to −0.166). However, BV decreased significantly from T1 to T2 (*p* < 0.001) and increased from T2 to T3 (*p* < 0.001). This trend was similar when using both mV_TG_ and pV_TG_, although the change in BV using mV_TG_ was greater than that using pV_TG_ at T1–T2 (*p* = 0.006; 95% CI = 0.063–0.257) and at T2–T3 (*p* = 0.001; 95% CI = −0.257 to −0.069). Db, calculated using mV_TG_, was increased significantly from T1 to T2 (*p* = 0.01) and remained unchanged from T2 to T3. However, when calculated using pV_TG_, Db increased significantly from T1 to T2 (*p* = 0.001) and decreased from T2 to T3 (*p* = 0.025). As a result, the change in Db using mV_TG_ was significant, but slightly lower than that using pV_TG_ from T1 to T2 (*p* = 0.007, 95% CI = −0.005 to −0.001) and from T2 to T3 (*p* = 0.002; 95% CI = 0.001–0.003).

To compare FM and FFM calculated using mV_TG_ and pV_TG_ in the 2C and 4C models, we used factorial repeated measures ANOVA with “time” and “methods” as independent variables (Figure 2). A statistically significant interaction (time × methods) indicated differences in FM (*p* < 0.001) and FFM (*p* < 0.01). No difference in FM was found between the T1 and the T2 for 4C model using mV_TG_ and at the T3, this was lower than at the T1 and T2. A Bland−Altman plot showed that no systematic error or bias was observed in FM using mV_TG_ and FM using pV_TG_, regardless of which model was used (Figure 3). The changes in FM and FFM are presented in Figure 4. The change in the FM with 2C model using pV_TG_ was significantly greater than that with 4C model using mV_TG_ during T1–T2 and T2–T3. However, the change in FFM with 2C model using pV_TG_ was significantly lower than that in 4C model using pV_TG_ during T1–T2 and T2–T3.

## 4. Discussion

The present study showed that the measured V_TG_ significantly increased after RWL and decreased after WR. This finding was consistent with our original hypothesis. The changes in Db after RWL and WR were different between the values obtained from calculations using mV_TG_ and those using pV_TG_. However, the difference in the change in Db had no effect on estimations of FM, FFM and %FM while the changes in the FM and FFM calculated using mV_TG_ did not differ from those obtained using pV_TG_ in either the 2C model or the 4C model.

Although these changes in V_TG_ had no significant influence on the difference in FM change, the difference in FM when using mV_TG_ and pV_TG_ in the 2C model was 0.7 ± 1.6 kg after RWL and 0.5 ± 1.5 kg after WR. On the other hand, the difference in FM using mV_TG_ and pV_TG_ in the 4C model was 0.3 ± 0.8 kg after RWL and 0.2 ± 0.7 kg after WR. Thus, the difference was smaller in the 4C model than in the 2C model. As a result, comparing the variation of FM and FFM with 4C model using mV_TG_ as a reference, the 2C model using pV_TG_ provided significantly different values. We believe that the cause of this effect was associated with the principle of ADP. The basic principle of ADP is based on the assumption that the density of the FM is 0.9 g/cm^3^ and that of the FFM is 1.1 g/cm^3^. These assumptions are based on the fact that the proportion of water in the FFM is 73% [27]. However, since our participants restricted their water intake during RWL, the proportion of water in the FFM was slightly decreased (72.5 ± 0.7% at T1, 71.9 ± 0.6% at T2, 72.7 ± 0.7% at T3). Because these values were slightly lower than the assumed value of 73% at any time point, the 2C model showed an overall lower average value than the 4C model and also slightly overestimated the change in the fat mass. Therefore, when measuring changes in body composition, it would be more accurate to apply the Db calculated using mV_TG_ and the 4C model, which takes body water content into consideration.

We observed that the measured V_TG_ increased after RWL before decreasing after 13 h of WR. Minderico et al. [12] previously reported that the measured V_TG_ increased by 0.20 ± 0.36 L after 3.5 ± 5.5 kg of weight loss in overweight and obese women. In the current study, because the participants lost more weight than those included in the study of Minderico et al. [12], further increases in V_TG_ were reasonable. The authors of this previous study considered that the changes in V_TG_ could be explained by the changes in abdominal fat and the improved redistribution of blood into the thoracic compartment, which increases vital capacity. The baseline FM values of the participants of the current study were much lower and the decrease in FM was lower than that observed in the study by Minderico et al. [12]. Therefore, we could not conclude that a reduction in abdominal fat contributed to the 0.4-L change in V_TG_ observed in this present study.

Another possible factor responsible for the observed increase in V_TG_ after RWL is the loss of visceral tissues. Although the effects of RWL on visceral organs were not directly measured in the present study, previous studies showed that organ mass decreased after a long-term weight loss program in overweight and obese subjects with or without type 2 diabetes [28,29,30]. Kukidome et al. [13] further reported that when wrestlers lost approximately 7.3% of their BM over the course of approximately 1 week, the abdominal cross-sectional visceral area on the Jacoby line decreased by ~25.8%, which was measured by magnetic resonance imaging. The reduction of liver or gastrointestinal mass has also been reported after three days of fasting in rodents [31,32]. The food weight and energy intake that our participants consumed during the RWL period was very low. Therefore, similar to previous studies [13,28,29,30,31,32], the volume of the internal organs decreased with RWL, leading to an expansion of the thoracic cavity and thus, it could be inferred that V_TG_ increased. The decrease in V_TG_ after WR (in contrast to RWL) may be the result of the rapid increase in the volume of the liver and other visceral tissues. Tai et al. [33] reported that the liver regained volume after 16 h of re-feeding following a 16% loss of BM in rodents. In a human study, Decombaz et al. [34] reported that the liver volume increased after ingesting a fructose or galactose drink in a 6.5 h recovery period after exercise. In the present study, participants consumed 4669 ± 1051 g and 12.10 ± 1.23 MJ during the first 5 h post-RWL. Post-WR body composition was measured approximately 7.5 h afterwards. At this time point, the substances consumed were still inside the gastrointestinal tract and could therefore expand the volume of some organs. This expansion would be enhanced by a reduction in digestive and absorptive functions after RWL as suggested by a previous study, which showed that total parenteral nutrition leads to small intestinal atrophy [35] and reduces blood flow throughout the intestinal tract [36]. In the present study, we did not measure visceral volume and function. However, we inferred that the RWL diminished both visceral tissue volume and functions, such as digestion and absorption, which may have affected the post-WR reduction in V_TG_.

RWL was found to induce some physiological variations, such as a decrease in plasma volume and blood flow [4] associated with dehydration; a decrease in ventilation and respiratory efficiency during exercise [37]; and decrease in fat free mass [5], strength, anaerobic power and anaerobic capacity [38]. It is considered that these changes induce a decrease rather than an increase in V_TG_. Another physiological parameter that is possibly affected by RWL is the increase in body temperature. In another study performed in our laboratory, when we examined the changes in body temperature at about the same time points as in this study, it was 0.3 ± 0.3 °C higher in the evening than in the morning with or without RWL. However, it was reported that a body temperature rise of 0.3 °C does not affect V_TG_ and body composition [39]. Although we did not analyze the actual body temperature because we did not actually measure body temperature in this study, the association of body temperature with V_TG_ changes cannot be excluded.

It has been shown that ADP overestimates the reduction in FM during RWL [40]. In a previous study, we examined the difference in FM and FFM variation using mV_TG_ or pV_TG_ with 4C model as the standard. As a result, although a statistically significant difference was not observed in these parameters, the Bland−Altman analysis revealed that the 95% CI fluctuated widely in the 2C model compared to the 4C model. However, ADP is convenient and widely used to measure body composition in athletes, particularly in weight-categorized sports that require weight control. Therefore, abandoning ADP appears to be impractical. In the cases where body weight changes rapidly, it should be noted that ADP using pV_TG_ to evaluate body composition changes can overestimate FM changes while body water changes should also be taken into account.

Our current study had limitations that should be considered when interpreting the results. First, we included elite collegiate wrestlers and it was difficult to find comparable controls. Second, the sample size was small and all participants exhibited the same range of RWL and WR. Third, the protocol of this study cannot be generalized as it addresses a specific group of subjects who aim to temporarily reduce and regain weight to participate in the competition and win, unlike weight loss in the general population. We speculate that the results of this study may be expanded to other specific weight-change populations (e.g., subjects before and after childbirth or subjects that undergo bariatric surgery). Finally, the factors affecting the change in the variation of mV_TG_ could not be completely understood. Thus, further studies should objectively examine the effect of RWL on visceral organs using magnetic resonance imaging to explain the changes in mV_TG_.

## 5. Conclusions

ADP-measured V_TG_ was increased by RWL, but was restored by WR. However, the changes in the FM and FFM obtained when Db was calculated using mV_TG_ and pV_TG_ did not differ in either the 2C or 4C model. We demonstrated that the use of the 4C model improved the results due to the inclusion of TBW when determining FM and FFM. We compared the 2C model with the 4C model mV_TG_, which was defined as the standard model for body composition assessment. When estimating the changes in body composition, it is desirable to evaluate them with the 4C model, which includes body water content in the model. Furthermore, it was suggested that the measured V_TG_ should be used during rapid weight loss and regain because if the predicted V_TG_ is used, the fat mass change is further overestimated.

## Figures and Tables

**Figure 1 sports-07-00048-f001:**
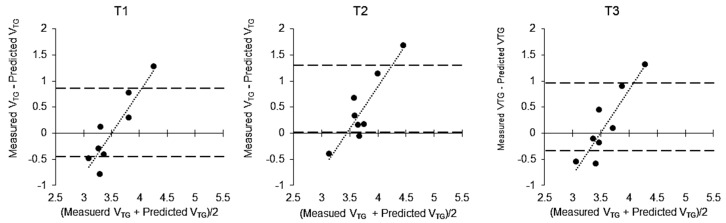
Bland−Altman plot of between-method differences in measured V_TG_ and predicted V_TG_ difference and the mean of V_TG_ and predicted V_TG_ at T1 (R^2^, 0.83, *p* < 0.01), T2 (R^2^, 0.81, *p* < 0.01) and T3 (R^2^, 0.80, *p* < 0.01). 95% confidence intervals are shown as dashed lines in the figure; V_TG_, thoracic gas volume.

**Figure 2 sports-07-00048-f002:**
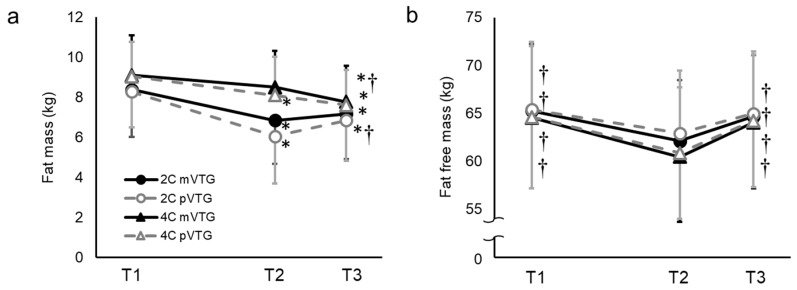
Fat mass (**a**) and fat free mass (**b**) during the experimental period (mean ± SD). BM, body mass; V_TG_, thoracic gas volume; RWL, rapid weight loss; WR, weight regain; WL; weight loss; Significant interactions between “time” and “methods” are indicated by repeated measures ANOVA. * *p* < 0.05 compared with T1, † *p* < 0.05 compared with T2 according to the Bonferroni post hoc test.

**Figure 3 sports-07-00048-f003:**
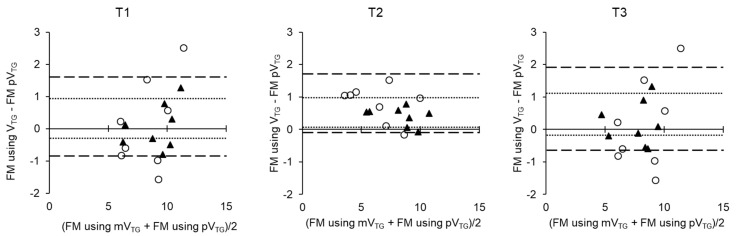
Bland−Altman plot of between-method FM using mV_TG_ and pV_TG_ differences versus mean of the two methods at T1, T2 and T3 by 2C model (○) and 4C model (▲) 95% confidence intervals of 2C model and 4C model are shown as dashed and dotted lines, respectively.

**Figure 4 sports-07-00048-f004:**
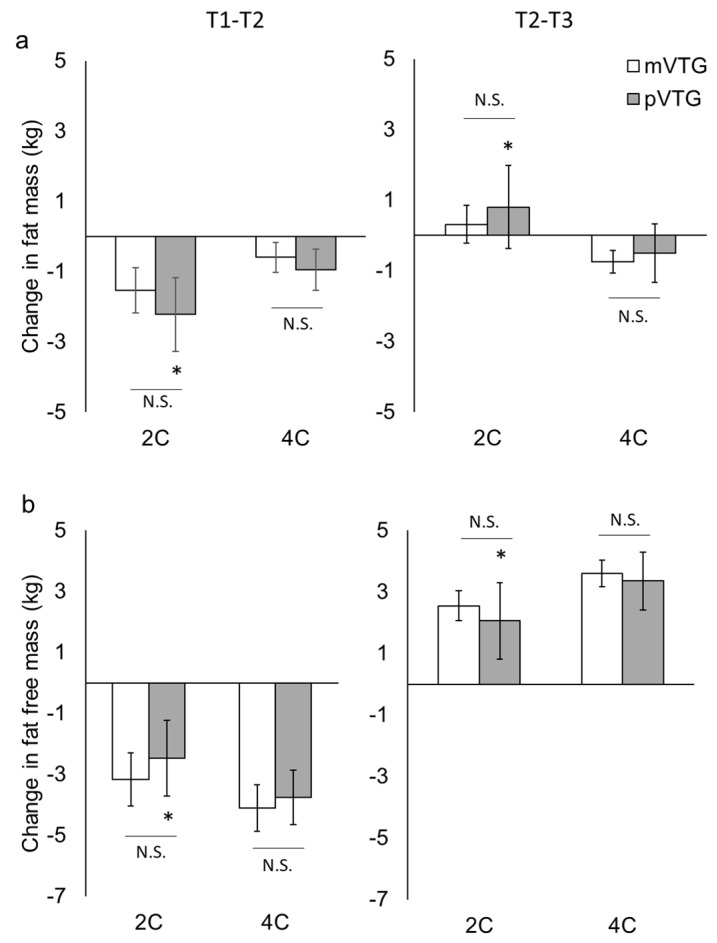
Changes in fat mass (**a**) and fat free mass (**b**) (mean ± SD). 2C, 2 component; 4C, 4 component; mV_TG_, measured V_TG_; pV_TG_, predicted V_TG_; Significant main effects (time) were obtained for fat mass and fat free mass in the 2C model and for fat free mass in the 4C model, as determined by repeated measures ANOVA. Furthermore, to compare with 4C model using measured V_TG_, one-way ANOVA and Bonferroni post hoc test were used. * *p* < 0.05 for the difference from 4C-model using mV_TG_.

**Table 1 sports-07-00048-t001:** Energy and macronutrient intake.

Variable	Unit	Baseline (Per Day)	RWL Period (53 h)	WR Period (13 h)
Food weight	(g)	3686 ± 1615	3234 ± 1591	4669 ± 1051
Energy	(MJ)	14.76 ± 3.47	9.90 ± 4.95	12.10 ± 1.23
	(kcal)	3528 ± 829	2366 ± 1184	2891 ± 295
Protein	(g)	125 ± 30	85 ± 41	64 ± 8
	(g/kg)	1.7 ± 0.4	1.2 ± 0.5	0.9 ± 0.0
	(%)	14.2 ± 0.9	14.6 ± 2.8	8.8 ± 0.2
Fat	(g)	110 ± 24	34 ± 20	60 ± 9
	(%)	28.9 ± 5.3	28.4 ± 9.3	18.5 ± 1.1
Carbohydrate	(g)	509± 149	340 ± 170	524 ± 45
	(g/kg)	6.9 ± 1.9	4.6 ± 2.1	7.1 ± 0.3
	(%)	56.9 ± 5.5	57.0 ± 11.4	72.7 ± 1.3

Data are expressed as means ± standard deviation (SD), n = 8; RWL, rapid weight loss; WR, weight regain.

**Table 2 sports-07-00048-t002:** Characteristics, body volume and body density of the participants.

Variables		T1	T2	T3	Change	
					T1–T2	T2–T3
Age		20.4 ± 0.5			±	±
Height (cm)		169.7 ± 3.5			±	±
BM (kg)		73.7 ± 8.0	69.0 ± 7.7 *	71.8 ± 7.7 *^#^	−4.7 ± 0.5	2.9 ± 0.3
TBW (kg)		46.8 ± 5.6	43.5 ± 5.2 *	46.6 ± 5.3 *^#^	−3.4 ± 0.6	3.1 ± 0.6
Mo (kg)		3.2 ± 0.3	3.2 ± 0.3 *	3.2 ± 0.3 *	0.0 ± 0.0	0.0 ±0.0
V_TG_ (L)	mV_TG_	3.56 ± 0.72	3.96 ± 0.70 *	3.67 ± 0.69 ^#^	0.36 ± 0.31	−0.29 ± 0.15 ^†^
	pV_TG_	3.51 ± 0.16	3.51 ± 0.16	3.51 ± 0.17	0.00 ± 0.01 ^a^	0.00 ± 0.01 ^a^
BV (L)	mV_TG_	68.67 ± 7.53	64.09 ± 7.26 *	66.76 ± 7.32 *^#^	−4.58 ± 0.43	2.66 ± 0.36 ^†^
	pV_TG_	68.65 ± 7.48	63.91 ± 7.22 *	66.69 ± 7.30 *^#^	−4.74 ± 0.46 ^a^	2.78 ± 0.35 ^†a^
Db (g/cm^3^)	mV_TG_	1.073 ± 0.006	1.076 ± 0.006 *	1.076 ± 0.006 *	0.004 ± 0.002	0.000 ± 0.002 ^†^
	pV_TG_	1.073 ± 0.005	1.080 ± 0.007 *	1.077 ± 0.006 *^#^	0.006 ± 0.003 ^a^	−0.002 ± 0.002 ^†a^

Data are expressed as means ± SD, n = 8; BM, body mass; V_TG_, thoracic gas volume; mV_TG_, measured V_TG_; pV_TG_, predicted V_TG_; TBW, total body water; Mo, Bone mineral content; BV, body volume; Db, body density; There were significant interactions for V_TG_, BV and Db: * *p* < 0.05 for the difference from T1, ^#^
*p* < 0.05 for the difference from T2, ^†^
*p* < 0.05 for the difference from T1–T2, ^a^
*p* < 0.05 for the difference from mV_TG_.

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
