# Peer review of "Effect of Thoracic Gas Volume Changes on Body Composition Assessed by Air Displacement Plethysmography after Rapid Weight Loss and Regain in Elite Collegiate Wrestlers"

_sports, 2019, doi:10.3390/sports7020048_

Round 1
Reviewer 1 Report
The authors present on an interesting study that assesses the changes in thoracic gas volume during rapid weight loss. An important consideration in the methods used is whether TGV is measured or predicted and the authors investigate the difference between the methods and indeed show different trends for TGV, which however do not translate to different fat mass estimates.
My major concerns about this study are
- lack of control group
- statistical comparison between two methods can be improved by using bland-altman-comparisons. These do not only allow for a direct estimate of error, but also of bias.
Minor concerns
- while I appreciate, that the likely change in hydration of fat-free mass is acknowledged in the discussion, this limitation of the 2C-model can be easily overcome, but calculation the 3C-model. Using Bland-Altman-Plots, the difference between the 2C, 3C and 4C models can be assessed, all using either mTGV or pTGV.
- it is mentioned that the athletes practiced TGV measurements, but was proficiency confirmed/defined? to this end, some test-retest-variability would be helpful and strengthen the meaningfulness of this analysis
- The introduction mentions the risk of such rapid weight loss, which may be associated with the observed dehydration? Variables of hydration (TBW/FFM) should be calculated, reported and discussed.
- The discussion on the effect of abdominal organ sizes on TGV seems speculative, and long for that matter. In that respect, I am wondering, what other factors might be involved? intercostal muscle tension/activity, dehydration, heart rate/stress, which may affect tidal volume rather than anatomical changes.
Author Response
The authors thank the reviewer for the time spent on examining the manuscript. The suggestions were very constructive, and we have extensively revised the entire manuscript based on these comments. The following is a point-by point consideration of these suggestions:
1. Lack of control group
Response: It was difficult to collect participants resembling comparable classes as a control group because we targeted elite collegiate wrestlers in this study. Therefore, we have added a measurement error of VTG data in a subgroup of athletes. Based on this comment, we have added the following sentence in the Methods section: “The intra-individual SD in a subgroup of athletes (n = 13; age, 21±3 years; height, 170.0 ± 5.7 cm) was 0.16 L for mVTG and 0.43 kg for FM for 2 consecutive days.” (Lines 140–142)
2. Statistical comparison between two methods can be improved by using bland-altman-comparisons. These do not only allow for a direct estimate of error, but also of bias.
Response: We thank the reviewer for this important suggestion. Based on this comment, the Bland-Altman plot could directly compare the two methods (measured VTG method and predicted VTG method), and further investigate their systematic errors. Thus, we have added the analysis and some figures. (Figure 1, Lines 220–222; Lines 251–252, Figure 3) As a result, there was a systematic error between pVTG and mVTG at all time points, but no systematic error was found in the calculated body composition by either the 2C model or 4C model. Therefore, we consider that the conclusion of this research was not affected by these data.
3. While I appreciate, that the likely change in hydration of fat-free mass is acknowledged in the discussion, this limitation of the 2C-model can be easily overcome, but calculation the 3C-model. Using Bland-Altman-Plots, the difference between the 2C, 3C and 4C models can be assessed, all using either mTGV or pTGV.
Response: We appreciate this further suggestion for 3C model; however, the main aim of this study was to confirm the accuracy of ADP (2C model) compared with the 4C model, which is regarded as the gold standard for body composition assessment. Thus, we have not assessed the 3C models to avoid confusion.
4. It is mentioned that the athletes practiced TGV measurements, but was proficiency confirmed/defined? to this end, some test-retest-variability would be helpful and strengthen the meaningfulness of this analysis
Response: We have described the inter-individual and intra-individual SD of five VTG measurements with the same participants (Lines 124–126). Additionally, we added the test-retest results according to Comment #1. (Lines 140–142) Therefore, we believe that these SD can be ensured the reliability of the measurement in this research.
5. The introduction mentions the risk of such rapid weight loss, which may be associated with the observed dehydration? Variables of hydration (TBW/FFM) should be calculated, reported and discussed.
Response: In accordance with this comment, we added the following information to the Discussion section: “However, since our participants restricted their water intake during RWL, the proportion of water in the FFM was slightly decreased (72.5 ± 0.7% at T1, 71.9 ± 0.6% at T2, 72.7 ± 0.7% at T3). Because these values were slightly lower than the assumed value of 73% at any time point, the 2C model showed an overall lower average value than the 4C model, and also slightly overestimated the change in the fat mass.” (Lines 272−276)
6. The discussion on the effect of abdominal organ sizes on TGV seems speculative, and long for that matter. In that respect, I am wondering, what other factors might be involved? intercostal muscle tension/activity, dehydration, heart rate/stress, which may affect tidal volume rather than anatomical changes.
Response: Thank you for your suggestion. In accordance with this comment, we have revised some information as follows: “RWL was found to induce some physiological variations, such as a decrease in plasma volume and blood flow (Reljic et al., 2013) associated with dehydration, a decrease in ventilation and respiratory efficiency during exercise (Caldwell et al., 1984), and a decrease in fat free mass (Sagayama et al., 2014), strength, anaerobic power, and anaerobic capacity (Webster et al., 1990). It is considered that these changes induce a decrease rather than an increase in VTG. Another physiological parameter possibly affected by RWL is the increase in body temperature. In another study performed in our laboratory, when we examined the changes in body temperature at about the same time points as in this study, it was 0.3 ± 0.3ºC higher in the evening than in the morning with or without RWL. However, it was reported that a body temperature rise of 0.3 ºC does not affect VTG and body composition. Although we did not analyze the actual body temperature because we did not actually measure the body temperature in this study, the association of body temperature with VTG changes cannot be excluded.” (Lines 321−331)

Reviewer 2 Report
The hypothesis of the present study is innovative that VTG would increase during RWL and decrease during WR what could lead to a difference in the estimation of body composition changes between values calculated with measured and predicted VTG. VTG is usually regarded to remain unchanged, even if BM changes the fixed VTG is usually used in sports science. These tests can change the current knowledge in the study of weight-categorized sports athletes who often reduce and regain their weight rapidly before match. For this reason, it is very important to accurately identify specific changes in their body composition to consider the strategy of weight management during severe rapid weight loss induces danger to health and life what must be controlled by confirmed methods.
The study is complete and well-designed, but should be useful or needed:
- to give, at the beginning, abbreviations used at work
- to underline the uniqueness of research
- the approval of the ethics committee (number) is needed; severe rapid weight loss is not safe for health and life
- to give detailed inclusion and exclusion criteria for group (e.g. years of training, amateurs, professionals, age, weight, etc.)
- in what period of the year the study was done; what method was used for selection of the study group (e.g. snowball method?), how representative is the group?
- Table 1 should be moved to results; my suggestion: to count the changes in sources of energy supply (baseline, RWL period, WR period) – how much energy was consumed from proteins, carbohydrates or fats (T1, T2, T3), the proportion was the same or changed; also the expression of energy intake in kcal would be useful for work.
Author Response
The authors thank the reviewer for the time spent to examine the manuscript. The suggestions were very constructive, and we have extensively revised the entire manuscript based on these comments. Below is included a point-by point consideration of these suggestions:
1. To give, at the beginning, abbreviations used at work
Response: We have spelled out abbreviations used for the first time in the abstract. (Lines 25–27)
2. To underline the uniqueness of research
Response: We have included the following information to underline the uniqueness of research in the conclusion: (Lines 352-360)
3. The approval of the ethics committee (number) is needed; severe rapid weight loss is not safe for health and life
Response: We have added the following information in the text of the manuscript: “approval number 036 in 2014” (Lines 90–91)
4. To give detailed inclusion and exclusion criteria for group (e.g. years of training, amateurs, professionals, age, weight, etc.)
Response: According with this comment, we have added the suggested information. (Lines 81–85)
5. In what period of the year the study was done; what method was used for selection of the study group (e.g. snowball method?), how representative is the group?
Response: According with this comment, we have added the information. “The study was conducted from January to March (winter season), and off-season of wresting. Subjects from two Kanto area teams in Japan were informed by their coaches, and those players who volunteered for cooperation and met the study criteria were recruited.” (Lines 79–81)
6. Table 1 should be moved to results; my suggestion: to count the changes in sources of energy supply (baseline, RWL period, WR period) – how much energy was consumed from proteins, carbohydrates or fats (T1, T2, T3), the proportion was the same or changed; also the expression of energy intake in kcal would be useful for work.
Response: In accordance to this suggestion, we have moved Table 1 to the Results section and revised it to Table 2 according to the appearance order. We also indicated the energy intake in kcal (Table 2).

Reviewer 3 Report
This is an interesting study with an appropriate design as body composition is an area of study that needs as much input, evaluating, and researching as possible, since pure anthropometry is impractical. This paper is well organized and well written. I genuinely hope that my comments will be construed as constructive and collegial, and I apologize for any instances in my review that appear to the contrary.
Specific comments:
The methods are explained in detail, however, can the authors confirm with a citation(s) that their protocol is common for RWL and WR in terms of timeframe. For example is 53 hours a standard amount of time engage in a RWL program leading up to a weigh-in? The same question goes to the 13 hours or WR.
Table 1: Please specify the time periods for baseline, RWL, and WR. Are baseline and RWL based on averages across several days and WR based on 5 hours?
Lines 280-283: The small sample size is noteworthy as a limitation, however no study is without multiple flaws. Suggest including information about not being able to objectively determine the affect RWL had on visceral organs which may very well explain the changes in mTGV. Also, this is a very homogenous sample with a specific goal and protocol to achieve the goals (RWL/WR) and that results cannot be generalized to other weight loss or weight gain populations. Please speculate on how future investigations can expand on what you have determined.
In the discussion, it
seems that the use of the 4C model may be very important due to the inclusion
of TBW when determining FM and FFM, however this topic did not receive much explanation
and how it may have a strong practical application in a population that
experiences fluctuations in TBW. The inclusion of the 4C model is a strength of this paper in addition to the attention given to the mTGV vs. pTGV. This information would fit around lines 274-275.
Author Response
The authors thank the reviewer for the time and effort spent to examine the manuscript. Several suggestions have strengthened the paper considerably and were incorporated in the manuscript. Authors provide below a point-by-point response to reviewer’s comments:
1. The methods are explained in detail, however, can the authors confirm with a citation(s) that their protocol is common for RWL and WR in terms of timeframe. For example is 53 hours a standard amount of time engage in a RWL program leading up to a weigh-in? The same question goes to the 13 hours or WR.
Response: To the best of authors’ knowledge, no clear definition of rapid weight loss is currently available; however, rapid weight loss is generally considered a reduction in body mass by 5% or more within a week. In this study, a period of 2.2 days (53 h) was adopted as weight loss period since most respondents indicated a period of 2 to 4 days in a previous weight loss survey at the All Japan Wrestling Championship (Kukidome et al. 2006 in Japanese). Furthermore, we referenced the protocol (59 h) and adjusted it according to the time schedule of this research. We added these previous studies to the reference list and cited them in the Methods section. (Line 93)
2. Table 1: Please specify the time periods for baseline, RWL, and WR. Are baseline and RWL based on averages across several days and WR based on 5 hours?
Response: In accordance with this comment, we described each period of energy and macronutrients intake. “At baseline, it was calculated based on 3-day food records adjusted by an intake per day for 6 training days and 1 training-off day, and the total intake in RWL period and WR period were 53 h and 5 h, respectively.” (Table 2 and Lines 174–176)
3. Lines 280-283: The small sample size is noteworthy as a limitation, however no study is without multiple flaws. Suggest including information about not being able to objectively determine the affect RWL had on visceral organs which may very well explain the changes in mTGV.
Response: As the reviewer suggested, we added the following information to the limitation section: “The effected variation of mVTG could not be completely understood. Thus, further studies should examine objectively the effect of RWL on visceral organs using magnetic resonance imaging to explain the changes in mVTG.” (Lines 348–350)
4. Also, this is a very homogenous sample with a specific goal and protocol to achieve the goals (RWL/WR) and that results cannot be generalized to other weight loss or weight gain populations. Please speculate on how future investigations can expand on what you have determined.
Response: The authors agree that the protocol of this study cannot be generalized, as it addresses a specific group of subjects that aim to temporarily reduce and regain weight to participate in the competition and win, unlike the weight loss in general population. However, the results of this study clearly showed that although the VTG may change in situations where body mass fluctuates greatly, like in weight loss or weight gain, it did not have a big influence on body composition estimation. We believe this is a very important point in assessing and monitoring the body composition of athletes who want to lose or gain weight and to change their body composition. According to the reviewer’s suggestion, we added a sentence to speculate on how future investigations can expand the results of the present study: “We speculate that the results of this study may be expanded to other specific weight-change populations (e.g., subjects before and after childbirth, or subjects that undergo bariatric surgery)” (Lines 346–348).
5. In the discussion, it seems that the use of the 4C model may be very important due to the inclusion of TBW when determining FM and FFM, however this topic did not receive much explanation and how it may have a strong practical application in a population that experiences fluctuations in TBW. The inclusion of the 4C model is a strength of this paper in addition to the attention given to the mTGV vs. pTGV. This information would fit around lines 274-275.
Response: We expected that this comment would probably have been suggested for the “conclusion” section. It is certainly a strength of this paper that we used the 4C model. Therefore, we have added the result of the Bland-Altman analysis and investigated the bias of the body composition evaluation 2C model and 4C model using mVTG and pVTG (Figure 3). We have also revised a portion of the conclusion. (Lines 352–360)

Round 2
Reviewer 1 Report
The reviewer appreciates the additional remarks, most importantly, the intra-individual SD's and the Bland-Altmans.